# Salvage Hepatectomy for Recurrent Hepatocellular Carcinoma after Radiofrequency Ablation: A Retrospective Cohort Study with Propensity Score-Matched Analysis

**DOI:** 10.3390/cancers15194745

**Published:** 2023-09-27

**Authors:** Yeshong Park, Ho-Seong Han, Yoo-Seok Yoon, Chang Jin Yoon, Hae Won Lee, Boram Lee, MeeYoung Kang, Jinju Kim, Jai Young Cho

**Affiliations:** 1Department of Surgery, Seoul National University Bundang Hospital, Seoul National University College of Medicine, Seongnam-si 13620, Gyeonggi-do, Republic of Korea; yeshong.park@gmail.com (Y.P.); yoonys@snubh.org (Y.-S.Y.); 82750@snubh.org (J.K.); 2Department of Radiology, Seoul National University Bundang Hospital, Seoul National University College of Medicine, Seongnam-si 13620, Gyeonggi-do, Republic of Korea

**Keywords:** hepatocellular carcinoma, HCC, radiofrequency ablation, RFA, liver resection, salvage hepatectomy

## Abstract

**Simple Summary:**

We evaluated the outcomes of salvage hepatectomy for local recurrence of hepatocellular carcinoma after radiofrequency ablation. Short-term outcomes including operation time, intraoperative blood loss, postoperative hospital stay, and postoperative complication rates were similar in salvage hepatectomy patients and a propensity score-matched control group who underwent liver resection as primary treatment. Recurrent tumors after radiofrequency ablation showed poorer differentiation, more aggressive behavior, and higher recurrence rates. Less extensive resection compared to the initial plan, negative but close (<0.1 cm) resection margin, and R1 resection were significant predictors for recurrence after salvage hepatectomy. These results suggest that salvage hepatectomy could be a rescue therapy for local recurrence after ablation, and wide resection margins are essential to prevent recurrence after surgery.

**Abstract:**

Background and Objectives: Although radiofrequency ablation (RFA) is a well-established locoregional treatment modality for hepatocellular carcinoma (HCC), the optimal strategy to handle local recurrence after ablation is still debated. This study aims to investigate the role of salvage hepatectomy (SH) as a rescue therapy for recurrent HCC after RFA. Materials and Methods: Between January 2004 and December 2020, 1161 patients were subject to surgical resection for HCC. Among them, 47 patients who underwent SH for local recurrence after ablation were retrospectively analyzed and compared to a propensity score-matched group of controls (n = 47) who received primary hepatectomy (PH). Short-term and long-term outcomes were analyzed between the two groups. Results: After matching, operation time, intraoperative blood loss, postoperative hospital stay, and postoperative morbidity rates showed no statistically significant difference. Tumors in the SH group were associated with poor differentiation (SH 9 (19.1%) vs. PH 1 (2.1%), *p* < 0.001). The 5-year disease-free survival rates (31.6% vs. 73.4%, *p* < 0.001) and overall survival rates (80.3% vs. 94.2%, *p* = 0.047) were significantly lower in the SH group. In multivariable analysis, less extensive resection compared to the initial plan (hazard ratio (HR) 4.68, *p* = 0.024), higher grade (HR 5.38, *P* < 0.001), negative but close (<0.1 cm) resection margin (HR 22.14, *p* = 0.007), and R1 resection (HR 3.13, *p* = 0.006) were significant predictors for recurrence. Conclusions: SH for recurrent tumors after ablation showed safety and effectiveness equivalent to primary resection. As recurrent tumors show a higher grade and more aggressive behavior, more extensive resections with wide surgical margins are necessary to prevent recurrence.

## 1. Introduction

Hepatocellular carcinoma (HCC) is the sixth most prevalent cancer worldwide, and its observed incidence rates are higher in East Asian countries, including Korea [1,2,3]. Therapeutic options are chosen based on several factors, including tumor number and size, liver functional reserve, and general performance status [4]. The Barcelona Clinic Liver Cancer staging system defines curative treatment options for early HCC as surgical resection, liver transplantation, and thermal ablation [5,6]. Local ablation techniques, including a percutaneous ethanol injection, microwave ablation, and radiofrequency ablation (RFA), are widely used in clinical practice and accepted as first-line therapy for early HCC cases [7,8,9]. Yet, HCC shows a limited long-term prognosis due to frequent recurrence, and local tumor progression rates after RFA have also been reported as high [4,10,11]. It is generally known that recurrence directly affects long-term survival in patients undergoing RFA [12].

Local recurrence after ablation is known to be associated with several risk factors, including tumor size, tumor number, and location near the liver surface [13]. When the complete ablation of the tumor is not achieved, patients are more likely to experience tumor progression; therefore, successful ablation without a residual tumor after ablation significantly improves survival [14,15]. Previous studies have also suggested that tumors exhibit more aggressive features after recurrence, possibly due to higher vascular invasion rates and dedifferentiation of the tumor via a heat shock effect during the thermal ablation procedure [10,16,17,18].

Repeated local ablation or transcatheter arterial chemoembolization (TACE) is mostly advocated as second-line treatment for local recurrence after initial ablation [19]. As recurrence after RFA is usually associated with advanced tumors, surgical resection could have a role as salvage therapy in selected patients. Yet, the existing literature on the role of surgical treatment after local recurrence is scarce, with most studies based on retrospective analyses with small sample sizes. This study aims to compare both the short-term and long-term outcomes of salvage hepatectomy for local recurrence after RFA with primary hepatic resection.

## 2. Materials and Methods

### 2.1. Study Population

We analyzed patients who underwent liver resection for HCC at a tertiary referral center in Korea (Seoul National University Bundang Hospital) between January 2004 and December 2020. Among the 1161 patients, those with repeated hepatic resection for recurrence after initial surgery (n = 214) and hepatectomy for recurrence after TACE (n = 353) were excluded from the analysis. As a result, a total of 594 patients were included, of whom 47 consecutive patients received salvage hepatic resection for local recurrence after RFA (SH group), and 547 patients underwent primary hepatectomy as their initial treatment (PH group). To adjust for clinicopathological differences between these two groups and reduce the potential effect of selection bias, 1:1 propensity score matching (PSM) was performed. Matching factors included patient demographics (age, sex), liver function (Child–Pugh class, liver cirrhosis), and tumor characteristics (number, size). Finally, 47 patients from the SH group and 47 patients from the PH group were successfully matched. This study was approved by the institutional review board of SNUBH and conducted in compliance with Strengthening the Reporting of Observational Studies in Epidemiology (STROBE) guidelines for cohort studies [20].

### 2.2. Data Collection and Definitions

Patient data, including baseline demographics, operative information, pathologic reports, and survival outcomes, were retrospectively collected from medical records. Hepatectomy procedures were categorized as described by the Brisbane classification [21]. Major liver resection was defined as the resection of three or more adjacent liver segments, such as hemihepatectomy or trisectionectomy. Minor liver resection included non-anatomical liver wedge resection, segmentectomy, or sectionectomy. Postoperative complications were graded according to the Clavien–Dindo system [22]. Mortality at any time during the patient’s postoperative hospital stay was defined as in-hospital death.

### 2.3. Surgical Procedures

Indications of salvage hepatectomy included a technical difficulty in repeating RFA due to tumor location or size, the presence of tumor thrombus, or individual patient preference. Whether major or minor hepatic resection could be performed was decided the same in the SH and PH groups based on preoperative imaging studies and liver functional reserve markers. In all study participants, macroscopically curative resection was planned. In the SH group, the initial computed tomography (CT) scans before RFA were additionally reviewed for the extent of resection necessary. This initial plan was compared to the actual hepatic resection performed after recurrence.

### 2.4. Follow-Up

All patients were assigned follow-up visits in the outpatient clinic at regular intervals. They underwent clinical examinations and screening for recurrence, including blood tests for the tumor markers α-fetoprotein (AFP) and des-γ-carboxyprothrombin (DCP) and imaging studies such as CT or magnetic resonance imaging. Recurrence was diagnosed when a hepatic lesion with radiologic features typical of HCC or distant metastasis was newly detected. The interval between the time of operation and the date of first recurrence was defined as disease-free survival (DFS). The interval between the time of operation and the date of cancer-related death or the last follow-up visit was defined as overall survival (OS). The median follow-up duration was 34 months.

### 2.5. Statistical Analysis

The Statistical Package for the Social Sciences (SPSS version 25.0, IBM Inc., Armonk, NY, USA) was used for statistical analyses. Normally distributed continuous variables were expressed as the mean (standard deviation), and non-normally distributed variables were presented as the median (interquartile range). Student’s *t*-test or the Mann–Whitney U test was used for comparisons. Categorical variables were indicated as the frequency (percentage); they were compared using Pearson’s chi-square test or Fisher’s exact test. Survival outcomes were compared using Kaplan–Meier analysis with lthe og-rank test. Risk factors for recurrence and cancer-related death were analyzed through Cox regression. All *p*-values were two-sided, and *p* < 0.05 was considered statistically significant.

## 3. Results

### 3.1. Study Population

Before matching, the SH and PH groups differed significantly in sex, tumor number, and tumor size. After matching, differences in baseline characteristics disappeared. The baseline characteristics of the patients before and after matching are summarized in Table 1.

### 3.2. Operative Parameters and Pathologic Features

After PSM, a comparison of operative parameters between the two groups showed similar tumor locations (Table 2). An operative approach, major hepatectomy and anatomical resection rates, operation time, and intraoperative blood loss showed no difference between the SH and PH groups. In the SH group, 31.9% of patients underwent a more extensive resection after recurrence compared to the initial plan; in 6.4%, a less extensive resection was performed.

A comparison of pathological features between the two groups showed no difference in tumor size and number (Table 3). The Edmonson–Steiner (ES) grade of the tumors was higher after salvage hepatectomy (grade III: SH 20 (42.6%) vs. PH 11 (23.4%), grade IV: SH 9 (19.1%) vs. PH 1 (2.1%), *p* < 0.001). There was no difference in either macrovascular or microvascular invasion rates. R1 resection rates were similar between the SH and PH groups (SH 1 [2.1%] vs. PH 3 [6.4%], *p* = 0.617).

### 3.3. Postoperative Outcomes

When postoperative outcomes were analyzed, postoperative morbidity rates showed no statistically significant difference between the two groups (Table 3). There was no mortality during hospitalization in either group. The duration of postoperative hospital stay was similar between the two groups (SH 7 [5–9] days vs. PH 7 [5–8] days, *p* = 0.683).

The cumulative overall 1-, 3-, and 5-year DFS rates of the whole study population were 70.6%, 62.9%, and 52.6%, respectively. Recurrence rates were higher after salvage hepatetomy for both local recurrence (SH 29 (61.7%) vs. PH 9 (19.1%), *p* < 0.001) and systemic recurrence (SH 17 (36.2%) vs. PH 3 (6.4%), *p* = 0.001). When 5-year DFS rates were compared, the outcomes were significantly worse for the SH group (SH 31.6% vs. PH 73.4%, *p* < 0.001). The cumulative overall 1-, 3-, and 5-year OS rates of the whole study population were 96.7%, 91.2%, and 87.7%, respectively. Cancer-related mortality rates were higher in the SH group (SH 12 (25.5%) vs. PH 3 (6.4%), *p* = 0.024). The 5-year OS rate was significantly lower in the SH group (SH 80.3% vs. PH 94.2%, *p* = 0.047). The survival curves are illustrated in Figure 1.

### 3.4. Regression Analysis for Risk Factors of Recurrence and Cancer-Related Death

Univariable and multivariable Cox regression analyses were performed for risk factors of recurrence in the SH group. In univariable analysis, less extensive resection compared to the initial plan, high tumor grade, negative but very close (less than 0.1 cm) surgical margin, and R1 resection were significant predictors (Table 4). In multivariable analysis, a less extensive resection compared to the initial plan (hazard ratio (HR) 4.68, *p* = 0.024), ES grade IV (HR 5.38, *p* < 0.001), negative but close surgical margin (HR 22.14, *p* = 0.007), and R1 resection (HR 3.13, *p* = 0.006) were all found to be significant. Subgroup analysis was performed for local and systemic recurrence. Risk factors for local recurrence included ES grade IV (HR 3.02, *p* = 0.010) and R1 resection (HR 6.20, *p* = 0.022). For systemic recurrence, only the poor differentiation of the tumor reached marginal significance as a predictor (HR 2.80, *p* = 0.057).

Univariable and multivariable regression was additionally performed for cancer-related death. In univariable analysis, the tumor grade and close resection margin predicted outcomes (Table 5). In multivariable analysis, both factors were found to be prognostic (ES grade IV: HR 10.97, *p* = 0.009; close resection margin: HR 68.53, *p* = 0.004). When patients underwent less extensive resection compared to the initial plan, surgical resection with a negative but close margin, or R1 resection were grouped together; their survival outcomes were significantly worse compared to patients who possessed none of these risk factors (Figure 2).

## 4. Discussion

RFA is considered a curative treatment option for early HCC, and it is actively considered when the functional reserve of a patient’s liver is limited [17,23]. Yet, recurrent HCC after RFA is associated with a higher malignant potential than primary tumors, and the treatment strategies for local recurrence and their long-term outcomes remain unexplored [13,24,25]. In the current study, we compared patients who received salvage resection for local recurrence after RFA with a control group undergoing primary liver resection for HCC. Operative parameters, including operation time, blood loss, and postoperative morbidities, showed no difference between the groups, which proved the safety and feasibility of salvage hepatectomy after RFA. However, the SH group showed significantly higher recurrence rates, which reflected the aggressive nature of recurrent tumors after RFA.

In certain cases, RFA procedures might render subsequent surgical procedures extremely difficult due to adhesion formation between the diaphragm, abdominal wall, and liver. For this reason, previous studies have questioned if salvage hepatectomy after RFA is technically feasible. Two studies have shown that salvage resection after RFA results in prolonged operation time, more bleeding, and higher concomitant extrahepatic resection rates [10,26]. Yamashita et al. also reported that they had to change the operative plan during surgery mainly due to extensive adhesions, which are secondary to RFA in the majority of salvage hepatectomy patients [27]. In our study, operation time, intraoperative blood loss, and postoperative morbidity rates showed no statistically significant difference. We found that 35.8% of patients undergoing salvage hepatectomy required resections with a wider extent compared to the estimated resection extent required before ablation. However, the minimally invasive approach, anatomical resection, and major hepatectomy rates were comparable to the PH group. In accordance with previous studies, there was no case of mortality during hospitalization in the SH group. According to these findings, we concluded salvage resection to be safe and feasible in local recurrence after RFA.

Only a few studies have reported on the long-term results of salvage hepatectomy after ablation. A Japanese study comparing salvage hepatectomy to primary hepatectomy reported no significant difference in OS; the survival rate of salvage hepatectomy was 67% at five years, which indicated a clear survival benefit [10]. However, this study reported high recurrence rates, with inferior DFS outcomes in the salvage group. Imai et al. reported an overall 5-year survival rate of 58.3% in the salvage hepatectomy group [28]. This survival outcome was comparable to the nationwide survey results for the surgical treatment of HCC in Japan [28]. By contrast, another study from Japan reported that salvage hepatectomy reached 5-year cumulative survival rates of only 9.5% [29]. Torzilli et al. found both disease-free and overall survival rates to be worse after salvage hepatectomy, with a 2-year OS of 44.4% [26]. This study included both primary liver tumors, such as HCC, and secondary metastatic liver lesions. In our study, OS rates after salvage and primary hepatectomy were equivalent. Long-term survival outcomes in the salvage group were superior to previous studies, with a 5-year OS of 78.0%. However, recurrence rates were significantly higher after salvage hepatectomy and systemic recurrence rates were especially high.

There is still ongoing debate on the pathophysiological mechanism underlying local recurrence after RFA. Three major hypotheses have been proposed regarding the recurrence mechanism after HCC local control therapy [30,31]. First, primary treatment failure with incomplete ablation might lead to early recurrence [13]. Recent studies have suggested that incomplete RFA might induce changes in the molecular phenotype, resulting in higher invasive and metastatic potential [32,33]. Sub-lethal heat shock conferred during ablation might lead to higher proliferation rates and increased chemoresistance [34]. Second, recurrence might arise from a preexisting microscopic tumor that is undetectable via imaging methods [35,36]. Lastly, HCC might recur due to RFA procedure-related causes, including the dissemination of malignant cells due to RFA needle-induced direct seeding, the transvenous spread from incompletely ablated lesions, or the microrupture of the tumor due to increased intratumoral pressure during ablation [37,38,39]. Previous studies have found that recurrent tumors show higher vascular invasion rates compared to primary tumors undergoing resection, which might be explained by the third hypothesis [10].

Yamamoto et al. found high proliferation, poor histological grade, and portal venous invasion as notable characteristics in recurrent HCC after RFA [29]. The findings of the current study were in accordance with the existing literature, as we found that tumors in the SH group showed a significantly higher ES grade. In multivariable analysis, a high tumor grade was an independent predictor for both recurrence and cancer-related death. These results support that tumor biology plays an essential role in the aggressive recurrence pattern of the SH group. On a molecular level, it was recently reported that tumors receiving RFA showed a higher expression of epithelial-mesenchymal transition-related genes and markers of tumor angiogenesis [40,41]. One study comparing the needle biopsy results before RFA and after ablation also found that the dedifferentiation of the tumor was observed after the procedure [16]. Ahmed et al. also reported that RFA stimulated extrahepatic tumor growth through the activation of a hepatocyte growth factor and vascular endothelial growth factor [42]. This could provide a potential explanation for the high rate of systemic recurrence in the SH group in the current study. Further prospective studies focusing on the alterations between molecular markers before and after ablation could further elucidate the underlying mechanism of such aggressive behavior.

Resection margin status is a known risk factor related to recurrence and cancer-related death after surgery for HCC [43]. Some studies have proved that wide surgical margins could decrease tumor recurrence; on the other hand, several others failed to show the survival benefit of more extensive resections [36,44,45,46,47]. As the oncologic significance of the resection margin width remains controversial, previous studies have proposed that certain HCC patient subgroups might benefit from wider margins, including those with high AFP levels, those undergoing non-anatomical resections, and those with microvascular invasion [43,48,49]. In the current study, negative but close (less than 0.1 cm) margins significantly impacted both DFS and OS in patients undergoing salvage hepatectomy. Recurrent tumors after RFA showed poor differentiation and more aggressive behavior, and previous studies have also reported higher vascular invasion rates. Therefore, it might be necessary to perform liver resection with wide resection margins for oncological safety in these cases.

Another finding of our study was that less extensive resection compared to the initial plan before RFA significantly increased the risk of recurrence. It was suggested by previous studies that viable tumor cells might be hidden in ablated lesions that seem like complete necrosis in imaging studies [50]. In a previous study, Portolani et al. suggested that salvage hepatectomy for local recurrence after RFA should encompass extensive resections to cover the necrotic area, which might hide active tumor cells [51]. Careful preoperative planning to a surgical extent with the consultation of imaging studies for the initial tumor before ablation is necessary to perform salvage hepatectomy effectively. Major hepatectomies encompassing both the recurrent lesion and the previously ablated area should be preferred if possible, considering the underlying liver condition of the patient.

This study has limitations. First, this was a single-center study conducted in a retrospective manner. Therefore, estimations for operative parameters and postoperative survival outcomes could only be made by comparing the study population to a matched control group. Future large-scale prospective studies might be helpful to further validate the outcomes of salvage hepatectomy. Second, the SH group included only those patients who were referred for surgical resection. They represented a limited proportion of all patients who experienced local recurrence after RFA, and it is highly possible that they are associated with less advanced tumors and better liver condition compared to patients who were found unsuitable for operative treatment. As a clear selection bias existed, our results might have underestimated the aggressive nature of recurrent HCC after RFA.

## 5. Conclusions

As short-term outcomes, including operation time, intraoperative blood loss, and postoperative complication rates, are comparable to primary hepatic resection, salvage hepatectomy is a viable rescue therapy for local recurrence after ablation. Recurrent HCC, after RFA, shows poor differentiation and exhibits more aggressive behavior, and wide resection margins are essential in salvage hepatectomy to prevent recurrence after surgery.

## Figures and Tables

**Figure 1 cancers-15-04745-f001:**
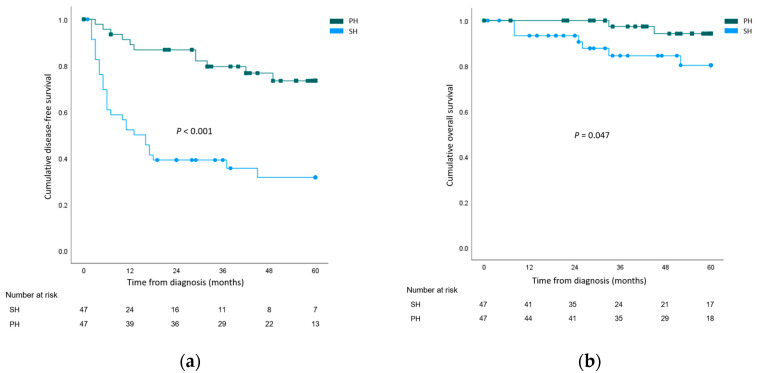
Survival analysis comparing the salvage hepatectomy (SH) group and the primary hepatectomy (PH) group. (**a**) Disease-free survival; (**b**) Overall survival.

**Figure 2 cancers-15-04745-f002:**
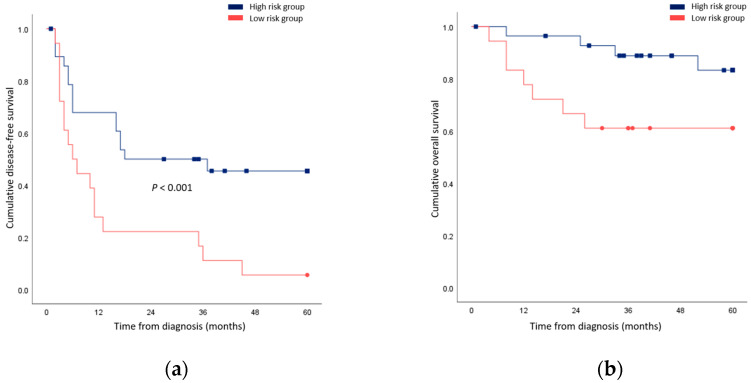
Comparison of survival rates between patients who underwent less extensive resection compared to the initial plan, a surgical resection with negative but close margin, or R1 resection (high risk group) and patients who possessed none of these risk factors (low risk group). (**a**) Disease-free survival; (**b**) Overall survival.

**Table 1 cancers-15-04745-t001:** Baseline characteristics of study participants before and after propensity score matching.

	Salvage Hepatectomy (n = 47)	Primary Hepatectomy				Total (n = 594)
		Before Matching (n = 547)	*p*-Value	After Matching (n = 47)	*p*-Value	
Age, years	60.9 (9.4)	60.3 (11.2)	0.722	59.2 (10.0)	0.390	60.4 (11.0)
Sex (male:female)	44:3	409:138	0.006	37:10	0.073	453:141
Hepatitis B	37 (78.7)	363 (66.4)	0.116	34 (72.3)	0.631	400 (67.3)
Hepatitis C	6 (12.8)	33 (6.0)	0.113	7 (14.9)	>0.999	39 (6.6)
Child–Pugh class			0.656		0.495	
A	45 (95.7)	530 (96.9)		47 (100)		575 (96.8)
B	2 (4.3)	17 (3.1)		0		19 (3.2)
MELD score	7.40 (6.76–8.42)	7.23 (6.54–8.00)	0.185	7.23 (6.87–8.23)	0.814	7.24 (6.54–8.09)
Platelet count, 10^3^/µL	171 (72)	184 (67)	0.226	181 (65)	0.492	183 (68)
Prothrombin time, INR	1.06 (1.01–1.10)	1.04 (0.99–1.10)	0.213	1.06 (1.03–1.10)	0.901	1.04 (1.00–1.10)
Total bilirubin, mg/dL	0.7 (0.6–0.9)	0.7 (0.5–1.0)	0.891	0.6 (0.5–0.9)	0.219	0.7 (0.6–1.0)
Albumin, g/dL	4.2 (0.4)	4.2 (0.5)	0.824	4.2 (0.4)	0.961	4.2 (0.5)
AFP, ng/mL	8.1 (3.2–59.3)	9.0 (3.3–76.1)	0.885	11.6 (3.6–122.7)	0.335	9.0 (3.3–70.8)
DCP, AU/mL	27 (16–97)	43 (20–261)	0.246	49 (19–307)	0.267	42 (19–246)
Liver cirrhosis	24 (51.1)	263 (48.1)	0.810	26 (55.3)	0.836	287 (48.3)
Preoperative tumor number	1 (1–2)	1 (1–1)	< 0.001	1 (1–1)	0.294	1 (1–1)
Preoperative tumor size, cm	2.9 (1.6–3.5)	3.0 (2.2–4.6)	0.027	3.0 (1.9–4.5)	0.170	3.0 (2.1–4.5)

MELD, Model for End-Stage Liver Disease score; INR, International Normalized Ratio; AFP, α-fetoprotein; DCP, des-γ-carboxyprothrombin. Values are presented as the mean ± standard deviation, median (interquartile range) or n (%).

**Table 2 cancers-15-04745-t002:** Operative parameters between patients undergoing salvage hepatectomy and primary hepatectomy.

	Salvage Hepatectomy (n = 47)	Primary Hepatectomy (n = 47)	Total (n = 94)	*p*-Value
Tumor location				0.312
Left	13 (27.7)	11 (23.4)	24 (25.5)	
Right anterior	13 (27.7)	20 (42.6)	33 (35.1)	
Right posterior	21 (44.7)	16 (34.0)	37 (39.4)	
Operative approach				0.661
Open	17 (36.2)	14 (29.8)	31 (33.0)	
Laparoscopic	30 (63.8)	33 (70.2)	63 (67.0)	
Operative extent				0.465
Major resection	13 (27.7)	9 (19.1)	22 (23.4)	
Minor resection	34 (72.3)	38 (80.9)	72 (76.6)	
Anatomical resection	21 (44.7)	20 (42.6)	41 (43.6)	>0.999
Deviation from initial plan				NA
More extensive resection	15 (31.9)	NA	NA	
Less extensive resection	3 (6.4)	NA	NA	
Operation time, min (mean ± SD)	238 ± 116	245 ± 120	241 ± 117	0.756
Pringle time, min	30 (15–56)	30 (20–40)	30 (19–41)	0.779
Intraoperative blood loss, mL	350 (300–900)	300 (150–700)	325 (200–700)	0.092
Intraoperative transfusion	7 (14.9)	6 (12.8)	13 (13.8)	>0.999

SD, standard deviation; NA, not applicable. Values are presented as the mean ± standard deviation, median (interquartile range) or n (%).

**Table 3 cancers-15-04745-t003:** Postoperative outcomes after salvage hepatectomy and primary hepatectomy.

	Salvage Hepatectomy (n = 47)	Primary Hepatectomy (n = 47)	Total (n = 94)	*p*-Value
Postoperative tumor number	1 (1–1)	1 (1–1)	1 (1–2)	0.635
Postoperative tumor size, cm	3.0 (2.2–4.7)	3.0 (1.9–4.5)	3.0 (1.9–4.5)	0.560
Edmonson and Steiner grade				<0.001
Grade I	0	6 (12.8)	6 (6.4)	
Grade II	18 (38.3)	29 (61.7)	47 (50.0)	
Grade III	20 (42.6)	11 (23.4)	31 (33.0)	
Grade IV	9 (19.1)	1 (2.1)	10 (10.6)	
Vascular invasion				
Macrovascular	5 (10.6)	4 (8.5)	9 (9.6)	>0.999
Microvascular	18 (38.3)	23 (48.9)	41 (43.6)	0.405
Margin status				0.617
R0	46 (97.9)	44 (93.6)	90 (95.7)	
R1	1 (2.1)	3 (6.4)	4 (4.3)	
Complication	9 (19.1)	10 (21.3)	19 (20.2)	>0.999
Atelectasis	0	2 (4.3)	2 (2.1)	
Pleural effusion	3 (6.4)	0	3 (3.2)	
Pulmonary thromboembolism	1 (2.1)	0	1 (1.1)	
Fluid collection	1 (2.1)	3 (6.4)	4 (4.3)	
Bile leakage	5 (10.6)	1 (2.1)	6 (6.4)	
Portal vein thrombosis	0	1 (2.1)	1 (1.1)	
Post-hepatectomy liver failure	1 (2.1)	0	1 (1.1)	
Urinary tract infection	0	1 (2.1)	1 (1.1)	
Ileus	1 (2.1)	0	1 (1.1)	
Wound complication	0	2 (4.3)	2 (2.1)	
C–D grade ≥ IIIa complication	6 (12.8)	4 (8.5)	10 (10.6)	0.738
Death during hospitalization	0	0	0	NA
Postoperative hospital stay, days	7 (5–9)	7 (5–8)	7 (5–9)	0.683
Recurrence				
Local recurrence	29 (61.7)	9 (19.1)	38 (40.4)	<0.001
Systemic recurrence	17 (36.2)	3 (6.4)	20 (21.3)	0.001
Cancer-related death	12 (25.5)	3 (6.4)	15 (16.0)	0.024

C–D, Clavien–Dindo; NA, not applicable. Values are presented as the median (interquartile range) or n (%) unless otherwise indicated.

**Table 4 cancers-15-04745-t004:** Univariable and multivariable regression analyses for recurrence after salvage hepatectomy.

	Univariable		Multivariable	
	HR (95% CI)	*p*-Value	HR (95% CI)	*p*-Value
Age, years				
≤60	Ref.			
>60	1.28 (0.61–2.71)	0.515		
AFP, ng/mL				
<200	Ref.			
≥200	1.06 (0.37–3.04)	0.916		
Hepatitis B infection				
No	Ref.			
Yes	0.66 (0.28–1.57)	0.35		
Hepatitis C infection				
No	Ref.			
Yes	1.87 (0.71–4.94)	0.209		
Operative approach				
Open	Ref.			
Laparoscopic	0.69 (0.33–1.42)	0.311		
Operative extent				
Minor resection	Ref.			
Major resection	1.30 (0.55–3.03)	0.551		
Anatomical resection				
No	Ref.			
Yes	0.82 (0.40–1.70)	0.598		
Deviation from initial plan				
No	Ref.		Ref.	
More extensive resection	0.88 (0.39–1.96)	0.750	0.84 (0.38–1.87)	0.673
Less extensive resection	5.04 (1.34–18.89)	0.017	4.68 (1.23–17.83)	0.024
Tumor number				
<2	Ref.			
≥2	1.40 (0.57–3.43)	0.463		
Tumor size, cm				
<3.0	Ref.			
≥3.0	1.53 (0.74–3.16)	0.249		
Tumor grade				
II/III	Ref.		Ref.	
IV	3.74 (1.67–8.37)	0.001	5.38 (2.22–13.03)	<0.001
Vascular invasion				
No	Ref.			
Yes	1.38 (0.67–2.85)	0.384		
Surgical margin				
Negative (>1 cm)	Ref.		Ref.	
Negative but close (≤1 cm)	8.75 (1.04–73.87)	0.046	22.14 (2.32–211.62)	0.007
Involved	2.27 (1.08–4.77)	0.031	3.13 (1.38–7.09)	0.006
Liver cirrhosis				
No	Ref.			
Yes	0.53 (0.25–1.12)	0.098		

HR, hazard ratio; CI, confidence interval; AFP, α-fetoprotein.

**Table 5 cancers-15-04745-t005:** Univariable and multivariable regression analyses for cancer-related death after salvage hepatectomy.

	Univariable		Multivariable	
	HR (95% CI)	*p*-Value	HR (95% CI)	*p*-Value
Age, years				
≤60	Ref.			
>60	1.44 (0.32–6.44)	0.637		
AFP, ng/mL				
<200	Ref.			
≥200	1.27 (0.15–10.58)	0.824		
Hepatitis B infection				
No	Ref.			
Yes	0.65 (0.13–3.36)	0.607		
Hepatitis C infection				
No	Ref.			
Yes	2.53 (0.49–13.04)	0.268		
Operative approach				
Open	Ref.			
Laparoscopic	0.69 (0.16–3.10)	0.630		
Extent of resection				
Minor resection	Ref.			
Major resection	1.06 (0.21–5.48)	0.943		
Anatomical resection				
No	Ref.			
Yes	1.54 (0.35–6.91)	0.570		
Deviation from initial plan				
No	Ref.			
More extensive resection	1.87 (0.38–9.29)	0.442		
Less extensive resection	3.57 (0.37–34.52)	0.272		
Tumor number				
<2	Ref.			
≥2	1.69 (0.33–8.74)	0.533		
Tumor size, cm				
<3.0	Ref.			
≥3.0	1.14 (0.25–5.12)	0.866		
Tumor grade				
II/III	Ref.		Ref.	
IV	6.88 (1.53–31.02)	0.012	10.97 (1.80–66.87)	0.009
Vascular invasion				
No	Ref.			
Yes	4.46 (0.87–23.02)	0.074		
Surgical margin				
Negative (>1 cm)	Ref.		Ref.	
Negative but close (≤1 cm)	18.73 (1.64–214.45)	0.019	68.53 (3.73–1260.91)	0.004
Involved	0.53 (0.06–4.57)	0.565	1.10 (0.11–11.18)	0.937
Liver cirrhosis				
No	Ref.			
Yes	2.14 (0.42–11.02)	0.364		

HR, hazard ratio; CI, confidence interval; AFP, α-fetoprotein.

## Data Availability

The data presented in this study are available on request from the corresponding author.

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
