# Peer review of "Salvage Hepatectomy for Recurrent Hepatocellular Carcinoma after Radiofrequency Ablation: A Retrospective Cohort Study with Propensity Score-Matched Analysis"

_cancers, 2023, doi:10.3390/cancers15194745_

Round 1
Reviewer 1 Report
The authors concluded that short-term outcomes of salvage hepatectomy for recurrent HCC after RFA is equivalent to that of primary resection. They also found that recurrent tumors showed higher grade and more aggressive behavior. The first half of this manuscript was described about the comparison between salvage hepatectomy and primary resection. The 5-year DFS and OS were significantly lower in the SH group than primary hepatectomy group. I do not understand why the authors concluded the short-term outcomes of SH is not different with that of primary resection. It might be meaningless to compare SH group with PH group because SH group was HCC recurrent group while PH group was not HCC recurrent group. Even if the propensity-score matched analysis was done, these groups were basically different. What they can say might be SH was safe and good therapy for recurrent HCC. This conclusion might be quite informative if this was correctly based on these results in this manuscript.
The latter half is well known to clinical physicians who had experiences of salvage hepatectomy. Risk factors of HCC recurrence after hepatectomy were not enough surgical margin and higher tumor grade as the authors explained.
It might be less interesting for hepatology specialist or investigators to read this manuscript.
Reviewer 2 Report
The series is from an experienced center with impressive numbers of resection for hepatocellular carcinoma (HCC). The authors have performed a propensity score matching of 47 patients who underwent salvage hepatectomy (SH) after radiofrequency ablation (RFA) and a similar number who underwent primary hepatectomy (PH) for HCC.
The difference in oncological outcomes between the SH and primary hepatectomy PH groups seems to be because of the aggressive tumors in the former group.
· Should not the two groups be matched for differentiation of tumor to compare the oncological efficacy of the treatment modality (RFA followed by SH versus PH)? For instance, comparison of disease free or overall survival in patients who had poorly differentiated HCC on histopathology after SH and PH
· Pre-operative parameters of tumor size, number, AFP and DCP do not seem to differ significantly from a clinical perspective
Minor point
Line 147 should be ‘no’ mortality
Reviewer 3 Report
The authors review the results of salvage hepatic resection for locally recurrent hepatocellular carcinoma after RFA. Compared to initial hepatectomy, they noted that postoperative results of salvage hepatectomy were poor, especially in patients with surgical margins of 1 cm or less.
Although this is a single-center study, the number of cases is relatively large, and the conclusions are convincing. Keeping sufficient resection margins in salvage liver resection will lead to improved outcomes, and this paper is considered to be useful in the surgical treatment of hepatocellular carcinoma.
Please respond to the following points
(i) The salvage liver resection group had more local recurrences, but also had more systemic recurrences. This is a problem that cannot be solved only by the issue of surgical margins, and please add a comment on this point.
(ii) Please prove that surgical margin is a significant factor related to local recurrence, even if the authors excluded the case with distant recurrence.
Round 2
Reviewer 1 Report
The manuscript has been improved and become informative.